# DNA methylation signatures of Alzheimer's disease neuropathology in the cortex are primarily driven by variation in non-neuronal cell-types

Gemma Shireby[1], Emma L. Dempster[1], Stefania Policicchio[1], Rebecca G. Smith[1], Ehsan Pishva [2], Barry Chioza [1], Jonathan P. Davies[1], Joe Burrage[1], Katie Lunnon [1], Dorothea Seiler Vellame [1], Seth Love[3], Alan Thomas[4], Keeley Brookes [5], Kevin Morgan [6], Paul Francis [1,7], Eilis Hannon [1] & Jonathan Mill [1] ✉

Alzheimer's disease (AD) is a chronic neurodegenerative disease characterized by the progressive accumulation of amyloid-beta and neurofibrillary tangles of tau in the neocortex. We profiled DNA methylation in two regions of the cortex from 631 donors, performing an epigenome-wide association study of multiple measures of AD neuropathology. We meta-analyzed our results with those from previous studies of DNA methylation in AD cortex (total *n* = 2013 donors), identifying 334 cortical differentially methylated positions (DMPs) associated with AD pathology including methylomic variation at loci not previously implicated in dementia. We subsequently profiled DNA methylation in NeuN+ (neuronal-enriched), SOX10+ (oligodendrocyte-enriched) and NeuN−/SOX10− (microglia- and astrocyte-enriched) nuclei, finding that the majority of DMPs identified in 'bulk' cortex tissue reflect DNA methylation differences occurring in non-neuronal cells. Our study highlights the power of utilizing multiple measures of neuropathology to identify epigenetic signatures of AD and the importance of characterizing disease-associated variation in purified cell-types.

Alzheimer's disease (AD) is a chronic and incurable neurodegenerative disease that is clinically characterized by progressive memory loss and declining cognition. Although AD is neuropathologically associated with the accumulation of extracellular amyloid-beta (Aβ) plaques and the deposit of intracellular neurofibrillary tangles of tau (NFT)[1,2], it is also frequently accompanied by pathological features associated with other types of dementia[3,4]. Lewy-body (LB) and TDP-43 pathology, for example, are often present alongside tau and amyloid pathology in individuals with AD[4]. Despite progress in identifying both genetic[5–9] and non-genetic risk factors for AD, the molecular mechanisms driving AD pathology remain elusive.

There is growing recognition of the importance of non-sequence-based regulatory variation in health and disease. Building on the hypothesis that epigenomic dysregulation is important in the etiology and progression of AD neuropathology[10], we and others have identified DNA methylation differences in several regions of the brain associated with AD and also other forms of dementia including Parkinson's disease (PD)[11–19]. A recent epigenome-wide association study (EWAS) meta-analysis, for example, reported >200 differentially methylated positions (DMPs) in the cortex associated with tau pathology[13]. There are, however, important limitations to existing studies of epigenetic variation in AD. First, because the cortex

comprises a heterogeneous mix of different cell-types—each characterized by a specific epigenetic signature—it is difficult to fully account for differences in cellular proportions between samples derived from "bulk" cortex tissue. Furthermore, because the progression of AD neuropathology is associated with changes in both the number and activation of specific cell-types—for example, AD is associated with both the loss of neurons[20,21] and the proliferation and activation of microglia[22,23]—studies performed on bulk cortex cannot identify disease-associated variation occurring within individual cellular populations. Second, the clinical and neuropathological heterogeneity among patients with AD, alongside the high level of comorbidity with other types of dementia, complicates the interpretation of associations between epigenetic variation and pathology. Although existing EWAS analyses of AD have largely focused on a single pathology measure (i.e., Braak NFT staging[1,24]), the simultaneous analysis of multiple measures of different types of pathology is likely to facilitate a better understanding of the molecular mechanisms involved in disease progression.

In this study we quantified genome-wide patterns of DNA methylation in the Brains for Dementia Research (BDR) cohort, a clinically and phenotypically well-characterized study established with the aim of integrating standardized measures of neuropathology with detailed phenotypic and multiomic data[25]. First, we performed a systematic EWAS of AD neuropathology, profiling DNA methylation across >800,000 sites in two cortical brain regions (the dorsolateral prefrontal cortex [DLPFC] and occipital cortex [OCC]) differentially impacted by AD pathology from ~650 well-characterized donors. Second, we meta-analyzed our results with those from previous AD EWAS analyses[13], enabling an analysis of AD-associated differential cortical DNA methylation in tissue from over 2000 individuals. Third, we characterized genome-wide patterns of DNA methylation in NeuN+ (neuronal-enriched), SOX10+ (oligodendrocyte-enriched), and NeuN–/SOX10– (microglia- and astrocyte-enriched) nuclei populations from a subset of BDR donors, exploring the extent to which AD-associated cortical differences in DNA methylation are driven by changes within specific cell populations. Our analyses identify neuropathology-associated variation at multiple novel loci not previously implicated in dementia and show that AD-associated methylomic variation in the cortex primarily reflects differences in non-neuronal cell populations. This study highlights the power of utilizing multiple neuropathology measures to understand the molecular pathogenesis of AD and the importance of characterizing disease-associated variation in distinct cell-types.

## Results

### An overview of the BDR DNA methylation dataset

After stringent data pre-processing and quality control filtering (see Methods), the final BDR dataset comprised of DNA methylation estimates for 800,916 DNA methylation sites profiled in 1221 tissue samples from two cortical brain regions (DLPFC and OCC) dissected from 631 donors (53% male, age range = 41–104 years, median age = 84 years, interquartile range [IQR] = 78–90 years, Table 1). Males were significantly younger at death compared to females (by 2.69 years, $P = 2.33E-07$), which is consistent with observations from epidemiological studies[26,27]. NFT pathology was quantified using Braak NFT staging[1,24] (mean Braak score = 3.72, SD = 1.90, Supplementary Fig. S1 and Table 1). Amyloid pathology was quantified using both Thal phase[2] (mean = 3.09, SD = 1.78) and neuritic plaque density scored using the CERAD classification method[28,29] (mean = 1.69, SD = 1.28). In addition, donors were also assessed for several hallmarks of non-AD pathology including both α-synuclein pathology using Braak LB staging[30] (mean = 1.34, SD = 2.26) and TDP-43 status (127 (22%) of 590 tested donors were classified as TDP-43 positive).

**Table 1 | Characteristics of the BDR samples profiled in this study**

| Analysis | Sample type | N | Age (years) | | Sex | | Neuropathology | | | | | | | | | |
| | | | Median (IQR) | Age range | Female | Male | Braak NFT stage | | CERAD score | | Thal phase | | LB stage | | TDP-43 status | |
| | | | | | | | N | Mean (SD) | N | Mean (SD) | N | Mean (SD) | N | Mean (SD) | N | Cases (%) |
| Bulk Cortex EWAS | Bulk cortex | 631 | 84 (78–90) | 41–104 | 296 | 335 | 618 | 3.72 (1.90) | 572 | 1.69 (1.28) | 558 | 3.09 (1.78) | 538 | 1.34 (2.26) | 590 | 127 (22) |
| Analysis of purified nuclei populations | Total nuclei | 26 | 80.5 (74–85.75) | 61–101 | 14 | 12 | 26 | 3.12 (2.47) | 24 | 1.46 (1.47) | 24 | 2.42 (1.90) | 22 | 0.723 (1.72) | 24 | 4 (17) |
| | NeuN+ | 27 | 81 (74–85.5) | 61–101 | 15 | 12 | 27 | 3.04 (2.46) | 25 | 1.4 (1.47) | 25 | 2.32 (1.93) | 23 | 0.696 (1.69) | 25 | 4 (16) |
| | SOX10+ | 28 | 80.5 (74–85.25) | 61–101 | 15 | 13 | 28 | 3.00 (2.41) | 26 | 1.35 (1.47) | 24 | 2.23 (1.95) | 24 | 0.791 (1.72) | 26 | 4 (15) |
| | NeuN–/SOX10– | 21 | 81 (74–85) | 61–101 | 13 | 8 | 21 | 3.19 (2.6) | 19 | 1.53 (1.47) | 19 | 2.47 (1.98) | 17 | 0.94 (1.92) | 20 | 3 (15) |
| Cell-deconvolution reference | NeuN+ | 12 | 81 (77.25–90.25) | 55–95 | 5 | 7 | 11 | 2.72 (1.74) | 10 | 1 (0.41) | 10 | 1.6 (0.97) | 10 | 0.4 (1.26) | 11 | 0 |
| | SOX10+ | 12 | 81 (77.25–90.25) | 55–95 | 5 | 7 | 11 | 2.72 (1.74) | 10 | 1 (0.41) | 10 | 1.6 (0.97) | 10 | 0.4 (1.26) | 11 | 0 |
| | NeuN–/SOX10– | 12 | 81 (77.25–90.25) | 55–95 | 5 | 7 | 11 | 2.72 (1.74) | 10 | 1 (0.41) | 10 | 1.6 (0.97) | 10 | 0.4 (1.26) | 11 | 0 |

*IQR interquartile range, SD standard deviation.*

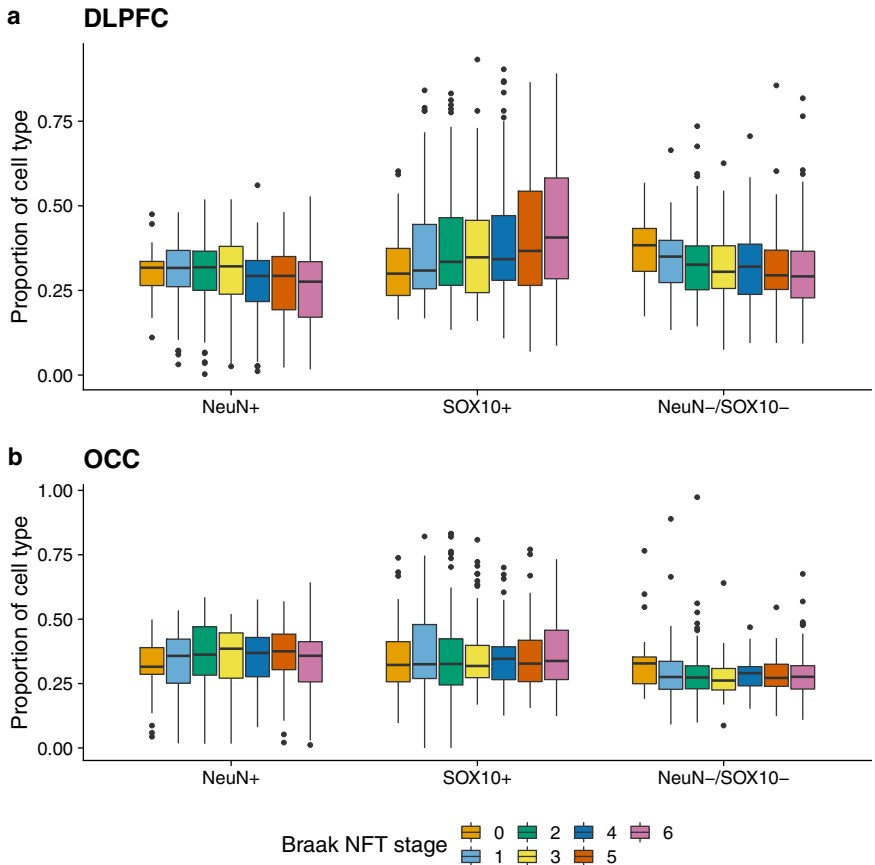

**Fig. 1 | Elevated tau pathology is associated with cell proportion estimates derived from DNA methylation data in the DLPFC but not the OCC.** Using linear regression models controlling for major covariates (see Methods) we show that **a** levels of tau pathology (measured using Braak NFT stage) are significantly associated with the proportion of NeuN+ cells (effect size = −2.74, SE = 0.705, *P* = 1.15E−04), SOX10+ cells (effect size = 1.60, SE = 0.423, *P* = 1.72E−04) and NeuN−/ SOX10− cells (effect size = −2.00, SE = 0.687, *P* = 0.004) in the DLPFC (*N* = 597 donors) using cell proportion estimates derived from "bulk" DNA methylation data. **b** In contrast no associations (*P* > 0.008) between levels of tau pathology and cell proportion estimates derived from "bulk" DNA methylation data were observed in the OCC (*N* = 598 donors). Boxplots of the estimated proportion of each cell-type across Braak NFT stages are shown, where the middle box represents the inter-quartile range (IQR), the middle line represents the median, and the whisker lines represent the minimum (quartile 1 −1.5 × IQR) and the maximum (quartile 3 + 1.5 × IQR). Tau pathology (Braak NFT stage) is shown on the *x*-axis split by cell-type and estimated cell proportions are shown on the *y*-axis. A similar pattern of results was found for levels of amyloid pathology as shown in Supplementary Fig. S2.

## Alzheimer's disease pathology is associated with altered cell-type proportions in the dorsolateral prefrontal cortex

The progression of AD pathology is associated with changes in the abundance of specific cell-types in the cortex; such changes in cell proportions are a major confounding factor for studies of DNA methylation and other genomic marks performed on "bulk" cortical tissue[31,32]. Although several methods have been developed to derive cell-type proportion estimates from bulk DNA methylation data for use as covariates in EWAS[31,33–36], these approaches are limited by the availability of DNA methylation reference data for specific cortical cell-types. We therefore used a fluorescence-activated nuclei sorting (FANS) method recently described by our group[37] to develop novel DNA methylation reference panels from NeuN+ (neuron-enriched), SOX10+ (oligodendrocyte-enriched), and NeuN−/SOX10− (microglia- and astrocyte-enriched) nuclei populations isolated from the DLPFC from a subset of control (low pathology) BDR donors (*n* = 12, see Methods and Table 1). DNA methylation profiles were generated from each purified nuclei population using the Illumina HumanMethylation EPIC microarray and used in combination with an established algorithm[31] to derive estimates for the proportion of each cell-type for all individuals included in our bulk cortex BDR datasets (see Methods). Of note, derived relative cell proportions were significantly associated with Braak NFT stage, CERAD score, and Thal phase in the DLPFC but not the OCC (see Fig. 1 and Supplementary Data 1), likely reflecting

known differences in the progression of neuropathology across the two brain regions. In the DLPFC, increasing tau pathology was significantly associated (Bonferroni *P* < 0.008 [0.05/6]) with reduced NeuN+ (neuronal) cell proportion estimates (effect size = −2.74; *P* = 0.00011), reduced NeuN−/SOX10− (microglial/astrocyte) proportions (effect size = −2.00; *P* = 0.004) and increased SOX10+ (oligodendrocyte) proportions (effect size = 1.60; *P* = 0.00017). This pattern was mirrored for the two measures of amyloid pathology (Supplementary Fig. S2 and Supplementary Data 1).

## Multiple differentially methylated positions were associated with AD neuropathology in the cortex

We used the detailed neuropathological data available for each BDR donor to identify cortical DMPs associated with the accumulation of both tau (measured by Braak NFT stage) and amyloid (measured by both CERAD score and Thal Phase) pathology. We first conducted an analysis of combined AD pathology incorporating all three AD pathology measures in a model including matched DLPFC and OCC DNA methylation data from individual donors that controlled for age, sex, derived cellular proportions, experimental batch, and principal component (PC) 1 (see Methods). We identified 67 DMPs annotated to 45 genes that were associated with the overall burden of core AD neuropathology at a stringent experiment-wide significance threshold (*P* < 9E−08) (Fig. 2a and Supplementary Data 2). Of note, 32 (48%) of

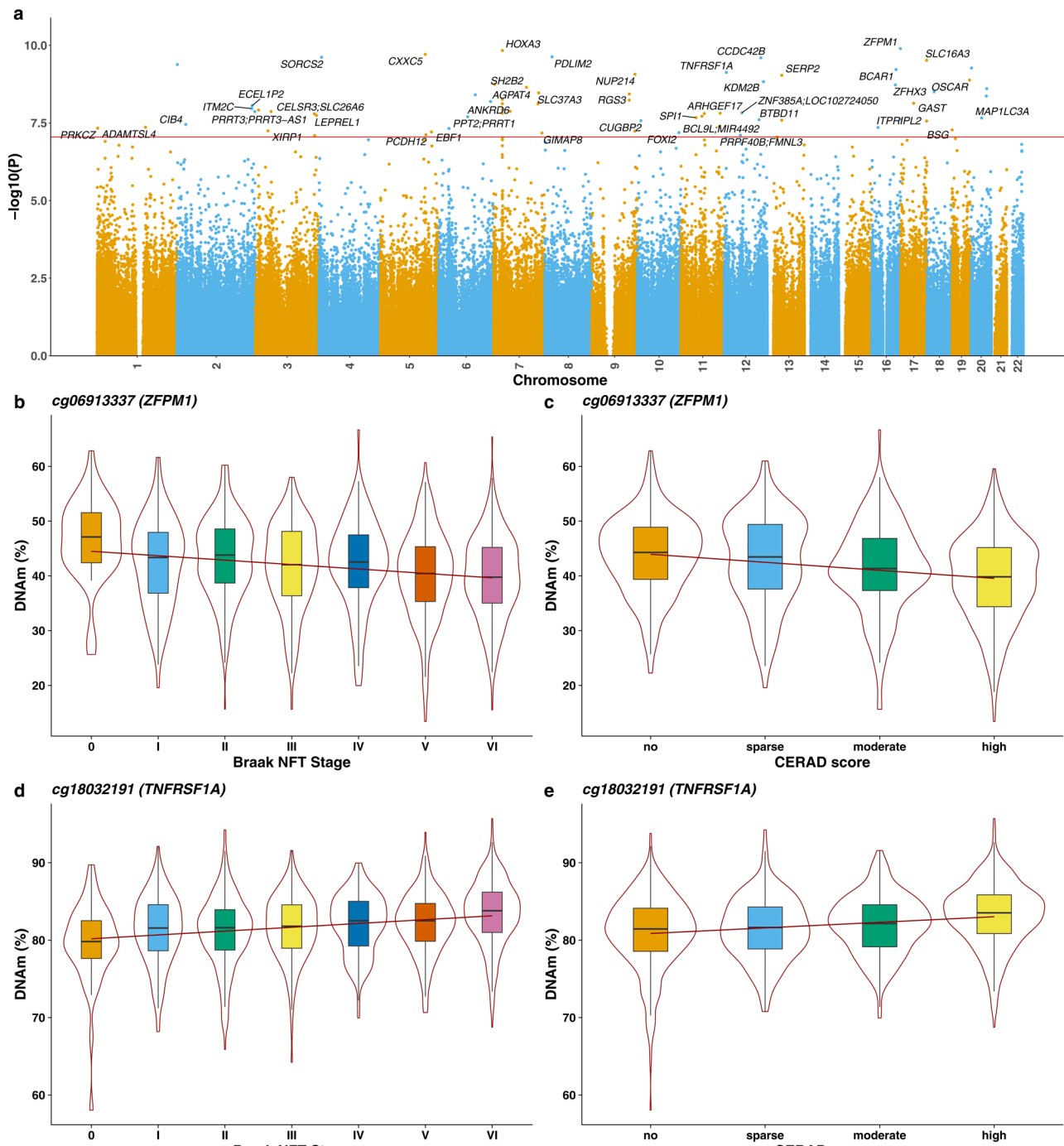

**Fig. 2 | Differentially methylated positions (DMPs) in the cortex associated with Alzheimer's disease neuropathology. a** Manhattan plot highlighting significant cortical DMPs associated with AD neuropathology (Braak NFT stage, CERAD score, Thal phase) (*N* = 631 donors). In total 67 DMPs associated with AD neuropathology were identified using linear regression models controlling for major covariates (see Methods) at an experiment-wide significance threshold (*P* < 9E−08). Genes annotated to significant DMPs are labeled. The *x*-axis depicts individual chromosomes 1–22 and the *y*-axis gives the significance level (−log10(*P*)) for each DNA methylation site tested. The horizontal red line represents the experiment-wide significance level (*P* < 9E−08). A complete list of results is given in Supplementary Data 3 and Manhattan plots showing results from EWAS analyses of individual AD neuropathology measures are given in Supplementary Figs. S3, S6, and S9. The top-ranked hypomethylated cortical DMP associated with AD neuropathology is

cg06913337 (annotated to *ZFPM1*). Lower DNA methylation at this site is significantly associated with **b** tau pathology (Braak NFT stage: effect size = −0.656%, SE = 0.0881%, *P* = 2.68E−09) and **c** amyloid pathology (CERAD score: effect size = −0.937%, SE = 0.162%, *P* = 6.64E−09). The top-ranked hypermethylated cortical DMP associated with AD neuropathology is cg18032191 (annotated to *TNFRSF1A*). Higher DNA methylation at this site is significantly associated with **d** tau pathology (Braak NFT stage: effect size = 0.322%, SE = 0.0598%, *P* = 7.20E−08) and **e** amyloid pathology (CERAD score: effect size = 0.46%, SE = 0.0893%, *P* = 2.53E−07). Shown are violin plots depicting DNA methylation values (adjusted for major covariates, see Methods) across pathology groups, where the box in the middle represents the interquartile range (IQR), the middle line represents the median and the whisker lines represent the minimum (quartile 1 −1.5 × IQR) and the maximum (quartile 3 + 1.5 × IQR).

the significant DMPs represent sites that are specific to the Illumina EPIC array and have not been assessed in previous analyses of AD cortex that have predominantly used the preceding Illumina 450 K array. The top-ranked cortical DMP associated with AD pathology was cg06913337, which was significantly hypomethylated with increasing AD pathology ($P = 1.27E-10$, Fig. 1b, c). Of note, this site is annotated to the *ZFPM1* gene that encodes a zinc finger protein that has been previously associated with DLB[38] and psychosis in AD[39].

## Differential methylation was associated with specific tau and amyloid pathology measures

We next undertook analyses to identify variable DNA methylation associated with each of the three individual AD pathology measures (Braak NFT stage, CERAD score, and Thal phase). First, we identified 26 DMPs annotated to 21 genes associated with tau pathology at an experiment-wide significance threshold ($P < 9E-08$) (Supplementary Fig. S3 and Supplementary Data 3). A total of 23 (88%) of these DMPs overlapped with sites identified in the AD neuropathology analysis. The average magnitude of effect per Braak NFT stage across these DMPs was 0.44% (SD = 0.17%), with a cumulative mean DNA methylation change of 2.63% (SD = 1.04%) from Braak stage 0–VI. Of note, 22 (83%) of the DMPs were significantly hypermethylated with a higher Braak NFT stage (enrichment $P = 0.000267$) reflecting the enrichment of hypermethylated loci observed in previous studies of tau pathology[13,16]. The top-ranked DMP (cg16021126) is annotated to *SERP2*, and was significantly hypermethylated with elevated Braak NFT stage ($P = 7.48E-10$, effect size = 0.29% per Braak NFT stage, Supplementary Fig. S4). *SERP2* is dysregulated in FTDP-17 (frontotemporal dementia and Parkinsonism linked to chromosome 17) iPSC-derived neurons[40]. A total of 16 (62%) of the 26 tau-associated DMPs identified in the BDR dataset were tested in a recent meta-analysis of tau pathology performed across sites on the Illumina 450 K array[13]; effect sizes for these sites were perfectly consistent across all tau-associated DMPs (100% concordant, binomial sign-test $P = 1.53E-05$, Supplementary Fig. S5a). It is notable that the magnitude of DNA methylation difference was approximately 2.2-fold larger in BDR than in the tau pathology meta-analysis (mean change per Braak NFT stage = 0.20% [SD = 0.09%]). Six (38%) of the 16 overlapping DMPs reached experiment-wide significance ($P < 9E-08$) in the previous meta-analysis and 14 (88%) reached Bonferroni significance correcting for 16 sites (Bonferroni $P = 0.00313$). Likewise, of the 220 DMPs identified in the tau pathology meta-analysis, 208 are included on the Illumina EPIC array and tested in the BDR dataset. These were characterized by highly consistent effect sizes observed across both analyses (100% concordant, binomial sign-test $P = 5.08E-61$, see Supplementary Fig. S5b); of note, effect sizes in the BDR cohort were again larger (average -1.2-fold larger) than those reported in the tau pathology meta-analysis.

Second, we identified 14 DMPs annotated to 12 genes associated with CERAD score (Supplementary Fig. S6 and Supplementary Data 3). The average magnitude of effect for the significant DMPs per unit of CERAD score was 0.57% (SD = 0.16%), with a cumulative absolute mean DNA methylation difference of 2.29% (SD = 0.63%) from low to high CERAD score and again an enrichment of hypermethylated sites (10 (71%) of DMPs showing higher DNA methylation with increasing pathology). The top-ranked DMP (cg13515047) is annotated to *BCAR1*, which encodes a Cas scaffolding protein that acts as a functional key regulator in the pathogenesis of AD[41], and was significantly hypermethylated with an elevated CERAD score ($P = 4.96E-09$, effect size = 0.44%, Supplementary Fig. S7 and Supplementary Data 3). Finally, we identified two experiment-wide significant DMPs associated with Thal phase, both hypermethylated with increasing pathology (Supplementary Fig. S8 and Supplementary Data 3). The top-ranked DMP (cg11658414, unannotated to any gene) was significantly hypermethylated with elevated Thal phase ($P = 9.11E-09$, effect size = 0.30%, Supplementary Fig. S9).

It is well established that the neuropathological signatures of AD are correlated and higher levels of NFTs are associated with elevated amyloid burden[42]. As expected, therefore, there was a strong positive correlation in patterns of differential DNA methylation across DMPs for the individual neuropathology measures assessed in BDR (see Supplementary Fig. S10). Effect sizes for the 26 Braak NFT stage DMPs, for example, were highly concordant (100%, binomial sign-test $P = 1.39E-17$ across all analyses) with effect sizes at the same DNA methylation sites in analyses of the other neuropathological measures in BDR (Supplementary Fig. S11). In addition, when fitting the full model controlling for all AD neuropathology measures, no DMPs remained significant ($P > 9E-08$) for each specific measure, further indicating the presence of common effects with consistent differences in DNA methylation across the different measures of AD neuropathology.

## Effect sizes at DMPs associated with AD pathology are correlated with those from an analysis of Lewy-body and TDP-43 pathology

Because other dementia neuropathologies are frequently present alongside tau and amyloid pathology in AD we sought to explore whether DNA methylation at AD-associated DMPs was associated with Braak LB stage and TDP-43 status, two measures of common co-pathology. The samples included in our study were characterized by limited amounts of both LB and TDP-43 pathology–the majority of donors were Braak LB Stage 0 ($n = 386$, 72%) and TDP-43 negative ($n = 463$, 78%)–and we therefore had limited power to identify novel DMPs associated with either type of pathology. Elevated TDP-43 status was associated with significant hypomethylation at a single DMP (cg06423355: $P = 5.47E-08$, effect size = −2.26%). Although this site is not directly annotated to any gene, it resides ~50 kb from *STK38L* that encodes a protein kinase involved in neuronal cell division and morphology and has been identified to control axonal growth in mouse hippocampal neurons[43]. Overall, effect sizes for the 67 AD pathology DMPs were found to be highly consistent between analyses of the AD (Braak NFT stage, CERAD score, Thal phase) and non-AD (Braak LB stage and TDP-43 status) neuropathology measures (see Supplementary Fig. S12) suggesting consistent effects across each type of neuropathology or that these effects are driven by underlying disease (i.e., a consequence rather than directly related to neuropathology).

## AD-associated differential DNA methylation is highly consistent across DLPFC and OCC

Our initial EWAS model leveraged matched DNA methylation data from both the DLPFC and OCC for each donor to maximize power to detect cortical DMPs associated with AD pathology. As expected, pathology-associated DNA methylation differences were highly consistent between both cortical regions across the 67 DMPs identified using this cross-cortex analysis model (binomial sign-test $P = 6.78E-21$, Supplementary Fig. S13). Given the progressive nature of AD pathology across different areas of the cortex, however, with more severe degeneration in the DLPFC compared to OCC[1,2,24]–as reflected in our finding of pathology-associated cell proportion changes in the DLPFC but not the OCC–it is plausible that there are brain region-specific differences in AD-associated patterns of DNA methylation. Therefore, we repeated our analysis including an interaction term for the brain region, identifying no significant region-specific associations with AD pathology ($P > 9E-08$). We also performed an EWAS of AD pathology (including the same three measures of tau and amyloid pathology) independently in each cortical region (Supplementary Data 4), identifying 30 significant DMPs in the DLPFC and 8 DMPs in the OCC (Supplementary Data 5 and 6). Although the larger number of DMPs identified in the DLPFC is consistent with the more advanced levels of AD pathology in this brain region compared to the OCC[1,2,24], effect sizes

were strongly concordant across regions (Supplementary Figs. S14 and S15) with one DMP (cg18100976, annotated to *PDLIM2*) being identified in both the DLPFC and OCC. Of note, *PDLIM2* encodes a protein that suppresses anchorage-dependent growth and promotes cell migration and adhesion, and has been implicated in PD by GWAS[44,45]. The consistency of findings between DLPFC and OCC suggests that variable DNA methylation at the identified DMPs is unlikely to simply reflect a consequence of neuropathology or neural cell loss.

### A meta-analysis of data from over 2000 donors identified over 300 cortical DMPs associated with tau pathology

We combined our BDR tau pathology EWAS results with the summary statistics from a recent analysis of tau pathology performed by our group[13], performing a cross-cortex inverse variance weighted (IVW) meta-analysis of the Braak NFT stage including data for 403,763 DNA methylation sites from 2013 donors derived from seven independent cohorts (the 6 cohorts included in the Smith et al. meta-analysis[13] in addition to the BDR samples described here (see Methods and Supplementary Data 7). In total, we identified 334 cortical DMPs (Bonferroni $P < 1.24E-07$) annotated to 171 genes (Fig. 3 and Supplementary Data 8). The full meta-analysis results for all probes tested are presented in Supplementary Data 9. Of note 140 (42% of the total) of these DMPs represented novel associations not previously identified in the previously published meta-analysis, reflecting the elevated power achieved by including the additional data from BDR donors. The top-ranked DMP, which was characterized by increasing DNA methylation with increased tau pathology (cg07061298: $P = 8.06E-18$, effect size = 0.32%, Fig. 3a) is annotated to *HOXA3*; of note, previous studies have strongly implicated differential DNA methylation across the HOXA region as being associated with AD pathology[13,46,47], and we found that 17 (5%) of the 334 meta-analysis DMPs are annotated to this genomic region (Supplementary Fig. S16). We also confirmed other previous AD EWAS associations, including a site annotated to *ANK1* (cg05066959; $P = 1.16E-13$, effect size = 0.41%) that has been robustly associated with AD pathology in previous EWAS studies of AD[11,15,16] and was characterized by elevated DNA methylation with increased tau pathology (Fig. 3b). Interestingly, several of the identified DMPs are annotated to genes that been also been implicated in GWAS analyses of AD pathology, including cg06784824 ($P = 1.71E-11$, effect size = 0.21%, Fig. 3c) annotated to *SPI1*, a gene hypothesized to regulate AD-associated genes in primary human microglia[7,48]. We performed gene ontology (GO) pathway analysis of the 171 genes annotated to the significant DMPs in the cross-cortex meta-analysis using *methylGSA* (see Methods) identifying significant enrichment of multiple pathways including pathways related to immune and inflammatory processes (see Supplementary Data 10 and Supplementary Fig. S17). Mounting evidence suggests the immune system plays a role in the etiology of AD and other dementias[49]; both local and peripheral inflammation is triggered by the degeneration of tissues (e.g., damaged neurons and neurites) and the deposition and highly insoluble proteins such as Aβ and NFTs[49]. Of particular interest was an enrichment of DMPs associated with genes involved in metalloproteinase activity pathways (e.g., "metalloendopeptidase activity" [GO: 0004222, $P = 5.09E-08$]); these proteins are important in neuroinflammation and have been strongly linked to neurodegenerative disease[50]. Other GO categories enriched amongst genes annotated to DMPs associated with tau pathology include pathways implicated in AD including several related to mitochondrial function (e.g., "mitochondrial transport" [GO: 0006839, $P = 5.09E-08$]) and "unfolded protein binding" (GO: 0051082, $P = 5.09E-08$). We subsequently repeated the meta-analysis focusing only on DLPFC samples from 1545 individuals from four independent cohorts (the 3 DLPFC cohorts included in the Smith et al. meta-analysis[13] in addition to the BDR DLPFC samples described here (see Methods and Supplementary Data 7), identifying 300 significant DMPs annotated to 161 genes (Supplementary Fig. S18 and

Supplementary Data 11). The full meta-analysis results for all probes tested are presented in Supplementary Data 12. There was considerable overlap between the results from both meta-analyses with 215 DMPs being significant in both, and the direction of effect being 100% concordant between the cross-cortex DMPs ($P = 2.86E-101$) and DLPFC DMPs ($P = 4.91E-91$) (Supplementary Fig. S19).

### An analysis of purified nuclei populations shows that the majority of DMPs identified in bulk cortex tissue reflect DNA methylation differences occurring in non-neuronal cells, with dramatically increased effect sizes observed in the NeuN−/SOX10− immunolabeled nuclei population

Although we attempted to control for potential heterogeneity in the proportion of different cell-types in our analysis of bulk cortex DNA methylation by using novel reference panels generated on NeuN+ (neuron-enriched), SOX10+ (oligodendrocyte-enriched), and NeuN−/SOX10− (microglia- and astrocyte-enriched) nuclei populations, our EWAS approach could not identify AD-associated differences occurring within specific cell populations. We therefore used our FANS protocol (see Methods) to profile DNA methylation in purified NeuN+, SOX10+, and NeuN−/SOX10− nuclei populations - in addition to a "total" nuclei population reflecting the cellular makeup of bulk cortex−from DLPFC tissue from a subset of "low" pathology (Braak score ≤II, $n = 15$) and "high" pathology (Braak score ≥V, $n = 13$) donors (Supplementary Data 13). We also co-stained nuclei with the microglial marker IRF8, highlighting complete overlap with the NeuN−/SOX10− population. Of note, a large proportion of NeuN−/SOX10− nuclei were IRF8+ (mean = 42.23%) indicating a relatively strong enrichment of microglia amongst this double-negative population. Of the DMPs identified in the DLPFC tau pathology EWAS meta-analysis, we obtained data for 327 sites in the purified nuclei populations ($n = 327$ DMPs). First, we looked at between-group effect sizes in the "total" nuclei population finding highly consistent DNA methylation differences to those seen in the large DLPFC meta-analysis despite the small number of samples, confirming the validity of our EWAS results (sign-test $P = 7.24E-46$, 87% concordant direction of effect). We then examined high vs low Braak score differences in DNA methylation at the 327 DLPFC DMPs finding a striking difference in the consistency and magnitude of effect sizes across each of the nuclei populations (Fig. 4). Although 67 DMPs (20%) had consistent directions of effects across all nuclei populations (Supplementary Data 14), the NeuN−/SOX10− population showed the most consistent between-group differences in DNA methylation (sign-test $P = 1.2E-75$, 96% concordant direction of effect) and was also characterized by a dramatic increase in effect sizes compared to those observed in bulk DLPFC (mean fold-change in effect size compared to bulk DLPFC = 10.72, Fig. 4). A similar pattern of differential DNA methylation was also observed in the SOX10+ (oligodendrocyte-enriched) population (sign-test $P = 2.15E-10$, 67% concordant direction of effect) again with an elevated effect sizes compared to bulk DLPFC, albeit to a lesser extent (mean fold-change in effect size compared to bulk DLPFC = 1.93, Fig. 4). These results suggest that the widespread cortical DNA methylation differences associated with AD neuropathology are primarily manifest in non-neuronal cell-types, although there is evidence for pathology-associated differences in cortical DNA methylation being specifically driven by variation in neuronal cell-types types for a subset ($n = 27$ (8.3%)) of tested DMPs (Supplementary Data 14).

### Discussion

We present a systematic analysis of cortical differences in DNA methylation associated with AD neuropathology. Using tissue and rich neuropathological data from 631 donors in the BDR cohort, we identified DMPs associated with levels of tau, amyloid, LB, and TDP-43 pathology across two cortical regions (DLPFC and OCC). We subsequently combined our results with those from previous studies of DNA methylation in AD cortex[13], performing a meta-analysis incorporating

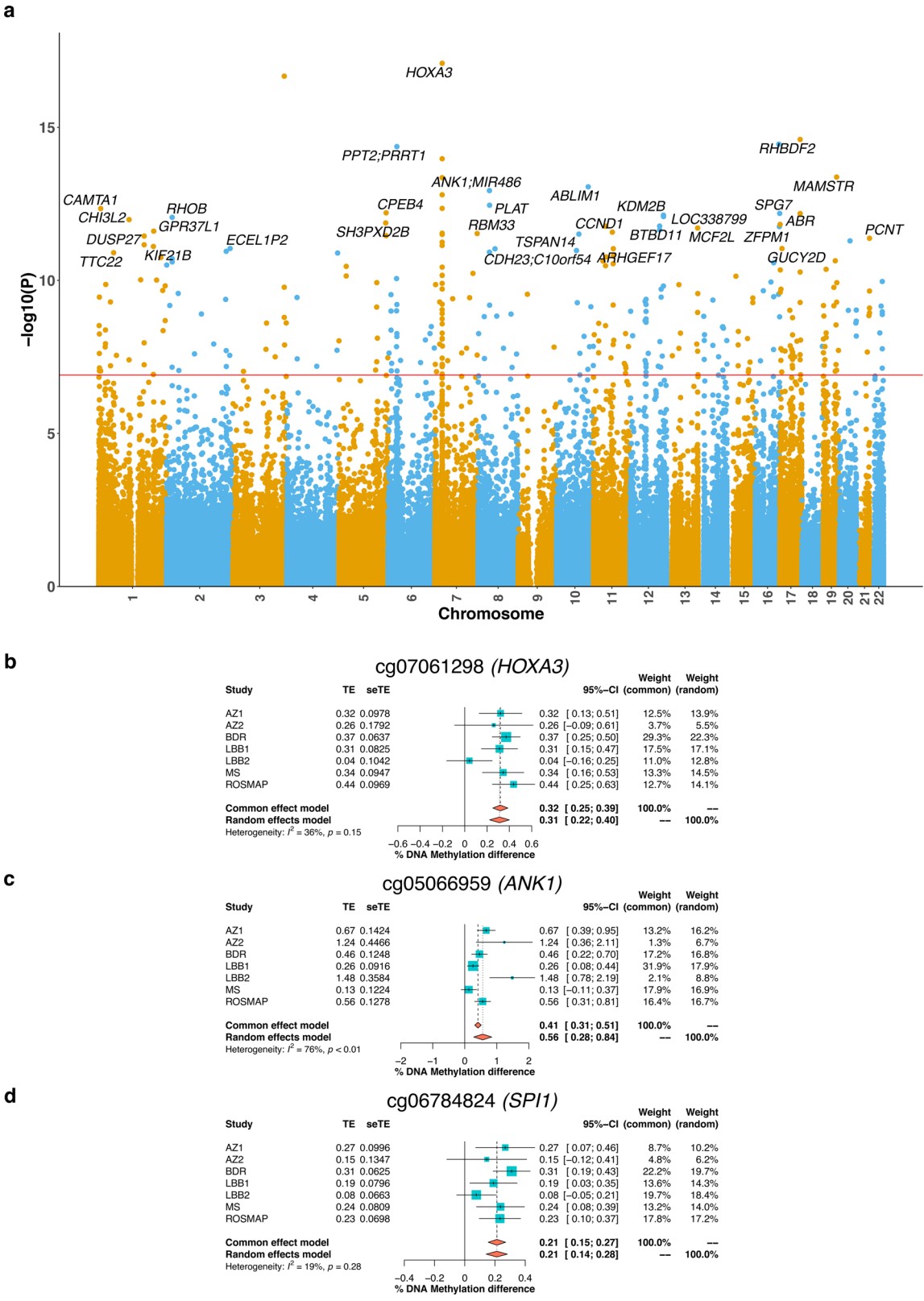

results from over 2000 donors and identifying 334 DMPs associated with AD pathology including many novel loci not previously identified in AD EWAS. We also characterized DNA methylation in purified immunolabeled DLPFC nuclei populations isolated from a subset of BDR donors with low and high AD pathology, exploring the extent to which pathology-associated DMPs are driven by differential DNA methylation in specific cell populations. Importantly, we find that the majority of DMPs identified in bulk cortex tissue reflect DNA methylation differences occurring in non-neuronal cells, with dramatically increased effect sizes observed in the NeuN−/SOX10− nuclei population. Our study highlights the power of utilizing multiple measures of neuropathology to identify epigenetic signatures of disease and the importance of characterizing disease-associated variation in purified cell-types.

**Fig. 3 | Differentially methylated positions (DMPs) identified in a cross-cortex meta-analysis include sites that are annotated to genes previously implicated in Alzheimer's disease. a** Manhattan plot highlighting significant cortical DMPs associated with Braak NFT Stage from an EWAS meta-analysis of all available AD datasets (total $N = 2013$ individuals). In total 334 DMPs associated with tau pathology were identified using linear regression models controlling for major covariates (see Methods) at an experiment-wide significance threshold ($P < 9E-08$). The x-axis depicts individual chromosomes 1–22 and the y-axis gives the significance level ($-log10(P)$) for each DNA methylation site tested. The horizontal red line represents the experiment-wide significance threshold ($P < 9E-08$). Gene annotations are given for the 50 top-ranked DMPs and a full list of results is given in Supplementary Data 9. Many of the DMPs associated with tau pathology have been previously implicated in AD. Elevated tau pathology is associated with **b** hypermethylation at cg07061298 (effect size = 0.32%, SE = 0.037%, $P = 8.06E-18$) that is annotated to *HOXA3* that has been implicated in previous EWAS analyses of AD pathology, **c** hypermethylation at cg05066959 (effect size = 0.41%, SE = 0.056%, $P = 1.16E-13$) that is annotated to *ANK1* that is also strongly implicated in previous EWAS analyses of AD pathology, and **d** hypermethylation at cg06784824 (effect size = 0.21%, SE = 0.032%, $P = 1.71E-11$) that is annotated to *SPI1* that is implicated in GWAS analyses of AD. The x-axis shows the effect size (% DNA methylation difference per SD increase in Braak NFT stage), with squares representing effect size and arms indicating the 95% confidence intervals. Details on each of the cohorts included in the meta-analysis (AZ1 Arizona 1, AZ2 Arizona 2, BDR Brains for Dementia Research, LBB1 London 1, LBB2 London 2, MS Mount Sinai, ROSMAP Religious Orders Study/Memory and Aging Project) are given in Supplementary Data 7.

Many of the pathology-associated DMPs identified in this study are annotated to genes that have previously been implicated in dementia. This includes multiple DMPs annotated to the HOXA region which has been previously identified in EWAS analyses of AD pathology[13,46,47]. The HOXA cluster is involved in the control of neuronal development, neuronal circuit organization, and the regulation of post-mitotic neurons[51,52], and in addition to AD methylomic variation across the HOX region has been associated with other neurodegenerative diseases including PD, Huntington's disease, and C9ORF72-related dementia[53–55]. AD pathology-associated DMPs were also annotated to many immune-related genes (e.g., *TNFRSF1A* and *OSCAR*) with GO pathway analyses finding an enrichment of immune and inflammatory pathways. These findings build on existing evidence that immune dysregulation plays a key role in the etiology of AD and other dementias[49]. In addition, differential DNA methylation in the vicinity of the *SPI1* gene was identified in our cortical meta-analysis of AD pathology. *SPI1* has been identified in recent AD GWAS[7,56] and EWAS[13] analyses and encodes the transcription factor PU.1, a pioneer factor for myeloid macrophages and microglial populations that has been implicated in regulating genes leading to inflammatory response in AD[48,57]. This is particularly interesting in the context of our analyses of sorted nuclei populations which identified that the majority of methylomic differences associated with AD pathology occur in the NeuN−/SOX10− population that is enriched for microglia.

The high overlap of DMPs and consistency of differences in DNA methylation across the different types of neuropathology assessed in BDR donors suggests that they may reflect some common signatures of neurodegeneration. This could imply that these differences are a common consequence of pathology or that they reflect the known pleiotropy between different types of dementia. For example, SNPs within the HLA region, MAPT, and APOE all contribute to increased risk for FTD, AD, and PD[58]. In addition, mutations in familial early-onset AD genes (*APP*, *PSEN1*, and *PSEN2*) are also observed in PD cases highlighting the pleiotropic effects associated with monogenic forms of neurodegeneration[59]. Previous EWAS analyses have also identified methylomic similarities between different neurodegenerative diseases[60] reporting significant over-representation in pathways related to brain function and immune response. The evidence for pleiotropy suggests that common pathological mechanisms likely underlie neurodegenerative disorders. Although neurodegenerative diseases differ in their neuropathological hallmarks and the specific brain regions involved, a common feature is the progressive accumulation of toxic protein deposits that ultimately lead to neuronal cell death and brain atrophy[61]. One key strength of the BDR dataset is that multiple neuropathology measures have been collected for each individual, enabling us to identify DMPs robustly associated with overall levels of AD neuropathology and leveraging greater power than analyses based on single pathology measures. Of note, although the findings suggest there are general methylomic signatures of neuropathological burden, we cannot exclude the presence of differential DNA methylation associated with specific types of neuropathology. Interestingly the

BDR effect sizes are larger than those observed in our recent meta-analysis of tau pathology[13]; this could potentially reflect cohort differences, the reduced heterogeneity in BDR, array platform differences, or by the fact that association statistics for variants meeting an experiment-wide threshold tend to be overestimated[62]. In addition, the consistency in the direction of effect demonstrates how robust the EWAS results for AD pathology are across studies.

A major strength of our study is our use of FANS to purify immunolabeled nuclei populations from a subset of donors prior to DNA methylation profiling. This enabled us to develop a refined cell-type deconvolution model that better controls for cellular heterogeneity in bulk cortex measurements of DNA methylation than previous models that only estimate the proportion of neuronal cells. Even when controlling for cell-type proportions, the bulk cortex analysis does not enable the identification of pathology-associated DNA methylation differences occurring in specific cell-types. We therefore profiled DNA methylation in FANS-purified nuclei populations from individuals with high and low AD pathology to explore the extent to which differences identified in bulk tissue were driven by variation in specific cell-types. Our analyses showed that most of the DMPs identified in the bulk cortex reflect variation in non-neuronal cell-types, with the biggest effect sizes identified in nuclei from NeuN−/SOX10− cells that we found to be relatively enriched for microglial (IRF8+) nuclei. These results support recent work highlighting a key role for microglia in AD[63]; with the activation of microglia colocalized with amyloid plaques in the brains of individuals with AD. The larger effect sizes observed at AD-associated DMPs in the microglial-enriched population might reflect the elevated reactivity of microglia in AD compared to other cell-types, presumably driven by cell-type-specific transcriptional signatures[63,64].

There are several limitations that should be considered when interpreting the results of this study. First, although we attempted to control for cellular heterogeneity and profiled FANS-purified populations to compare effect sizes across different cell-types, there are some limitations to this approach—for example, there is still considerable heterogeneity in each of the purified nuclei populations used to generate our deconvolution reference panels. The NeuN−/SOX10− fraction, for example, will comprise a mix of glial cell-types[65]; although co-staining with the microglial marker IRF8 highlighted that this population is relatively enriched for microglial cells (Supplementary Fig. 20) it will also include astrocytes and other non-neuronal cell-types. Furthermore, the use of NeuN as a marker to purify neuronal nuclei is not perfect[66]. Since neurodegenerative processes are associated with atrophy of astrocytes[65], they are an important cell-type to consider and future work should aim to further dissect the associations identified in the NeuN−/SOX10+ nuclei population. In the future, a reference dataset that includes astrocytes and other cell-types would also be optimal to more systematically control for cellular heterogeneity in bulk cortex DNA methylation data. Despite the relatively small number of purified nuclei samples profiled in our study, we were able to identify dramatically increased effect sizes in specific cell populations, highlighting

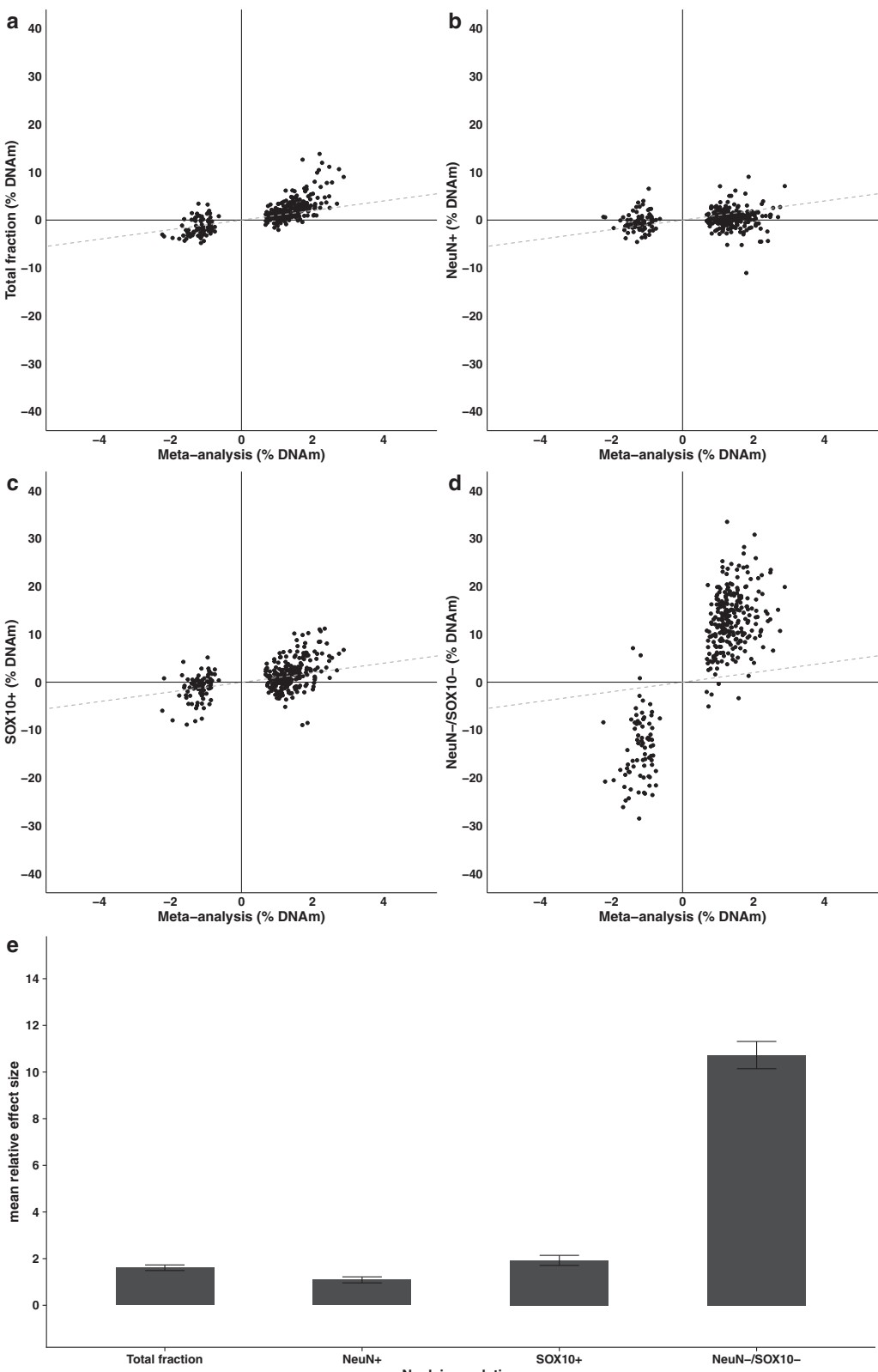

the additional power gained by profiling purified cell populations; further work in larger numbers of isolated nuclei populations is likely to yield even more striking evidence for cell-type-specific DNA methylation differences in AD pathology.

A key limitation of epigenetic studies of disease relates to the issue of causality; it is not possible to elucidate whether the DMPs identified in this study play a causal role in driving disease pathogenesis, whether they represent a direct downstream consequence of neuropathology, or whether they are induced by certain factors associated with AD pathology in the BDR cohort. In this regard, it is interesting that AD-associated differences identified in the OCC—a region of the cortex relatively protected from tau and amyloid pathology—were highly consistent with those identified in the DLPFC, which is affected much earlier in the disease process[1,2,24]. This consistency across both cortical

**Fig. 4 | Differentially methylated positions associated with AD pathology in the cortex largely reflect DNA methylation differences in non-neuronal cell-types.** We compared effect sizes for the 334 overlapping tau-associated DMPs identified in our "bulk" cortex meta-analysis with those at the same sites in an analysis of purified DLPFC nuclei populations from low (Braak NFT stage 0 to II) and high (Braak NFT stage >V) tau-pathology donors. Shown is a comparison of effect sizes between the meta-analysis (bulk, $N$ = 2013 individuals]) and the **a** total nuclei (bulk) nuclei fraction ($N$ = 26) (direction of effect = 87% concordant, sign-test $P$ = 7.24E−46); **b** NeuN+ (neuron-enriched) nuclei fraction ($N$ = 27) (direction of effect = 60% concordant, sign-test $P$ = 7.59E−05), **c** SOX10+ (oligodendrocyte-enriched) nuclei fraction ($N$ = 28) (direction of effect = 67% concordant, sign-test $P$ = 2.15E−10), and **d** double-negative (microglia- and astrocyte-enriched) nuclei population ($N$ = 21) (direction of effect = 96% concordant, sign-test $P$ = 1.2E−75). The $x$-axis shows effect sizes from the bulk cortex meta-analysis and the $y$-axis shows effect sizes for those same DMPs in each purified nuclei population. Gray dashed line represents $y = x$. **e** Bar-plots of the mean absolute relative effect sizes in each purified nuclei population compared to the bulk cortex across the 334 tau-associated DMPs, with error bars denoting the 95% confidence intervals.

regions suggests that the AD-associated variation identified in this study does not simply represent a consequence of AD neuropathology. Of note, however, we cannot exclude the possibility that the differences identified reflect the influence of other factors related to AD pathology that were not controlled for in this study, for example, environmental factors and other exposures such as medication that are themselves associated with AD pathology.

In summary, utilizing extensive neuropathology data from the BDR cohort we have performed a comprehensive EWAS of multiple measures of AD neuropathology across two regions of the cortex. Our meta-analysis with other AD DNA methylation datasets identified 334 cortical DMPs associated with AD pathology including methylomic variation at multiple loci not previously implicated in dementia. We subsequently characterized DNA methylation in purified nuclei populations finding that the majority of DMPs identified in bulk cortex tissue reflect DNA methylation differences occurring in non-neuronal cells, with increased effect sizes observed in SOX10+ and NeuN−/SOX10− nuclei populations. Our study highlights the power of utilizing multiple measures of neuropathology to understand epigenetic signatures of disease and the importance of characterizing disease-associated variation in purified cell-types.

## Methods

### The Brains for Dementia Research cohort
The BDR cohort was established in 2008 and represents a network of six dementia research centers across England and Wales (based at Bristol, Cardiff, King's College London, Manchester, Oxford, and Newcastle Universities) and five brain banks (brain donations from Cardiff are banked at King's College London)[25]. Briefly, participants >65 years of age were recruited using both national and local press (e.g., newspapers, newsletters, leaflets), TV and radio coverage as well as at memory clinics and support groups. There were no exclusion or inclusion criteria for individuals with specific diagnoses or those carrying genetic variants associated with neurodegenerative diseases; the cohort includes those with and without dementia and covers the full range of dementia diagnoses. Participants underwent a series of longitudinal cognitive and psychometric assessments and gave written informed consent for the use of tissue samples and clinical information for research purposes. Ethical approval for the study was granted by the University of Exeter Medical School Research Ethics Committee (13/02/009).

### Post-mortem neuropathological assessment of BDR brain donations
Post-mortem brain donations to BDR undergo full neuropathological dissection, sampling, and characterization by experienced neuropathologists in each of the five network brain banks using a standardized BDR protocol based on the BrainNet Europe initiative[67,68]. This protocol was used to generate a description of the regional pathology within the brain together with standardized scoring. Five variables representing four neuropathological features were used in the analyses presented in this paper: (1) Braak NFT stage that captures the progression of NFT pathology[1,24], (2) Thal phase that captures the regional distribution of Aβ plaques[2], (3) CERAD score that quantifies neuritic plaque density[29], (4) Braak LB stage that captures the

progression of α-synuclein throughout the brain[30,69], and (5) TDP-43 status—a binary indicator of the TDP-43 inclusions, which was assessed using immunohistochemistry to identify the presence of phosphorylated TDP-43 in the amygdala, hippocampus, and adjacent temporal cortex.Braak NFT stage, Thal phase, CERAD score, and Braak LB stage were analyzed as continuous variables, utilizing the semi-quantitative nature of these measures to identify dose-dependent relationships of increasing neuropathology with variable DNA methylation. TDP-43 status was analyzed as a binary variable.

### DNA methylation profiling in bulk cortex tissue
DNA methylation data were generated on two cortical regions (DLPFC and OCC) from each BDR donor. DNA was isolated from ~100 mg of tissue using the Qiagen AllPrep DNA/RNA 96 Kit (Qiagen, cat no.80311) following tissue disruption using BeadBug 1.5 mm Zirconium beads (Sigma Aldrich, cat. no. Z763799) in a 96-well Deep Well Plate (Fisher Scientific, cat. no. 12194162) shaking at 2500 rpm for 5 min. Genome-wide DNA methylation was profiled using the Illumina EPIC DNA methylation array (Illumina Inc.), which interrogates >850,000 DNA methylation sites across the genome[70]. After stringent data quality control (see below) the BDR dataset consisted of DNA methylation estimates for 800,916 DNA methylation sites profiled in 1221 samples (631 donors [53% male], 610 DLPFC, 611 OCC; age range = 41–104 years, median age = 84 years, mean age = 83.49 years, Table 1).

### Fluorescence-activated nuclei sorting of different cell populations from DLPFC
NeuN+, SOX10+, and NeuN−/SOX10− nuclei populations were isolated from ~700 mg of DLPFC tissue using a method optimized by our group[37]. First, nuclei populations were isolated from 12 donors with low neuropathology (Table 1) to generate reference DNA methylation profiles for purified nuclei populations for subsequent statistical deconvolution of brain cell proportions from bulk cortex DNA methylation data. Second, nuclei populations were isolated from DLPFC tissue from 15 low pathology (Braak score ≤II) and 13 high pathology (Braak score ≥V) BDR donors (total $N$ = 28, Table 1 and Supplementary Data 13) to identify cell-type-specific variable DNA methylation associated with AD pathology. A full protocol detailing each step of our nuclei purification protocol is provided at https://www.protocols.io/view/fluorescence-activated-nuclei-sorting-fans-on-huma-36wgq4965vk5/v1. Briefly, following tissue homogenization and nuclei purification using sucrose gradient centrifugation we used a FACS Aria III cell sorter (BD Biosciences) to simultaneously collect populations of NeuN+ (neuronal-enriched) (R&D systems, Cat No: NL2864R, dilution: 1:10) and SOX10+ (oligodendrocyte-enriched) (Millipore, Cat No: MAB377X, dilution: 1:1000) immunolabeled populations from bulk DLPFC tissue prior to genomic profiling, with the double-negative fraction and an aliquot of the "total" nuclei fraction (analogous to "bulk" cortex) also being collected from each tissue sample (Supplementary Fig. S20). In parallel, nuclei were also co-stained for IRF8 (Invitrogen, Cat No: 17-9852-82, dilution: 1:150), a microglial marker, to verify successful NeuN and SOX10 immunostaining and enable us to quantify the proportion of microglial nuclei in the NeuN−/SOX10− population. Nuclei suspensions were assessed for the presence of debris by adjusting the gating strategy before

proceeding with nuclei capture. For each sorted population, ~200,000 nuclei were collected for extraction of genomic DNA (Supplementary Data 13). Genomic DNA was isolated from each nuclei population using a standard phenol:chloroform extraction protocol[71] and DNA methylation was profiled using the Illumina EPIC array as described above.

## DNA methylation data pre-processing and quality control

Raw Illumina EPIC data were processed using the *wateRmelon* package as previously described[72]. Our stringent QC pipeline included the following steps: (1) checking methylated and unmethylated signal intensities and excluding poorly performing samples; (2) assessing the chemistry of the experiment by calculating a bisulphite conversion statistic for each sample, excluding samples with a conversion rate <80%; (3) identifying the fully methylated control sample included on each plate was in the correct location; (4) multidimensional scaling of sites on the X and Y chromosomes separately to confirm reported sex; (5) using the 59 SNP probes present on the Illumina EPIC array to confirm that matched samples from the same individual (but different brain regions or nuclei populations) were genetically identical and to check for sample duplications and mismatches; (6) using the *pfilter()* function in *wateRmelon* to exclude samples with >1% of probes with a detection $P$ value > 0.05 and probes with >1% of samples with detection $P$ value >0.05; (8) using PC analysis on data from each tissue to exclude outliers based on any of the first three PCs; and (9) the removal of cross-hybridizing and SNP probes[73]. The subsequent normalization of the DNA methylation data was performed using the *dasen()* function with the default options in either *wateRmelon* or *bigmelon*[72,74]. DNA methylation data generated on the purified nuclei populations were normalized separately for each cell-type.

## Identification of differential DNA methylation associated with neuropathology

To identify associations between variable DNA methylation and neuropathology we fitted regression models using the R (version 4.1) statistical environment[75]. As DNA methylation data for each donor was derived from two cortical regions, a mixed effects regression model was used, implemented with the *lme4*[76] and *lmerTest*[77] packages (see supplementary Fig. S21). To identify DNA methylation sites associated with AD neuropathology we conducted an EWAS in which DNA methylation at each probe was regressed against the three measures of tau and amyloid pathology (Braak NFT stage, CERAD density, and Thal Phase) using mixed effect regression models where age, sex, experimental batch, PC1 (which accounted for residual structure in the data) and derived brain cell proportions were included as fixed effects and individual was included as a random effect. Cell proportion estimates were derived from bulk cortex DNA methylation data using the Houseman method[31], implemented with *minfi* functions and default parameters, incorporating the novel reference DNA methylation data generated on three FANS-purified nuclei populations (NeuN+, SOX10+, and NeuN−/SOX10−) from 12 DLPFC samples (see Supplementary Fig. S20). Briefly, this method combines the cell-type reference data (generated from FANS-isolated nuclei populations) with bulk cortex data and performs quantile normalization. It then performs an ANOVA to identify sites that are significantly different ($P$ value <$1 \times 10^{-8}$) between the different cell-types and selects 100 sites per cell-type (50 hypermethylated and 50 hypomethylated). These sites are then used to derive cellular proportions using quadratic programming, in essence, a least squares minimization, with the constraint that all the proportions are greater than or equal to 0 and the sum of the three proportions is less than or equal to 1. Two of the three estimated proportions (NeuN+ and NeuN−/SOX10−) were included in the model to eliminate the effects of multicollinearity. To generate $P$ values, an ANOVA was conducted, comparing the full model including the three AD neuropathology measures to a null model in which the three measures were excluded.

We next conducted an EWAS for each of the five neuropathology measures separately (Braak NFT stage, CERAD score, Thal phase, Braak LB stage, and TDP-43-status) using the same set of covariates. In addition, we ran analyses where cell proportions were regressed against neuropathology in each brain region using linear regression models, controlling for age and sex. To identify tissue-specific effects, linear regression models were run in each brain region for the three main AD neuropathology measures controlling for age, sex, experimental batch, PC1, and derived cell proportions. Finally, to further explore if there was an effect present in one cortical region and not the other we ran a heterogeneity test, where we included an interaction between neuropathology and brain region in the mixed effects models, controlling for age, sex, experimental batch, brain region, PC1, derived cell proportions and individual. EWAS results were subsequently processed using the *bacon* R package[78], which applies a Bayesian method to adjust for inflation in EWAS.

## Meta-analysis of variable DNA methylation associated with AD pathology

Cross-cortex and DLPFC-specific meta-analyses of the Braak NFT stage were conducted incorporating the BDR with cohort-level summary statistics from a recent meta-analysis published by Smith et al.[13]; this paper includes detail on each of the individual cohorts included in the overall meta-analysis and information about each cohort is also provided in Supplementary Data 7. We first reran the Braak NFT stage EWAS in the BDR cohort excluding a small number ($N = 14$) samples that overlapped with samples included in the LBB1 cohort described in the Smith et al meta-analysis[13]. In the cross-cortex meta-analysis, a total of 2939 samples (from 2013 donors) were included (Supplementary Data 7). In the DLPFC meta-analysis, a total of 1,545 individuals were included (Supplementary Data 7). An IVW method was used that summarizes effect sizes from multiple independent studies by calculating the weighted mean of the effect sizes using the inverse of the variance of each study as weights. The EWAS results from each cohort were processed using the *bacon* R package[78]. A meta-analysis was then performed using the *metagen* function in the R package *meta*[79], using the effect sizes and standard errors from each individual cohort to calculate weighted pooled estimates and test for significance. Probes were limited to those present in at least two of the cohorts (cross-cortex $n = 403,763$ DNA methylation probes; DLPFC $n = 402,412$) and the $P$ value was Bonferroni corrected to control for this number of sites tested (cross-cortex $P < 0.05/403,763 = 1.24E{-}07$; DLPFC $P < 0.05/402,412 = 1.24E{-}07$). $P$ values are from two-sided tests and significant DMPs were taken from a fixed effects model. Pathway analyses were subsequently performed on the significant DMPs using the *methylglm* function within the *methylGSA* package developed by Ren and Kuan[80] using the default parameters.

## Regression against AD in FANS sorted nuclei populations

To determine whether associations identified in the bulk cortex are primarily driven by alterations in specific cell-types we used data generated on purified nuclei populations from individuals with high or low AD pathology. Briefly, we conducted an analysis of DNA methylation differences for significant sites from the bulk cortex meta-analyses comparing high and low pathology (defined as Braak high ≥V [$N = 13$]; Braak low ≤ II [$N = 15$]) (Braak score), which was modeled as a binary variable, in the four FANS sorted nuclei populations (total nuclei [analogous to "bulk" cortex], NeuN+, SOX10+, NeuN−/SOX10− separately. Linear regression models were used, whereby the significant DNA methylation sites identified in the cross-cortex and DLPFC meta-analysis were regressed against high/low pathology status controlling for age, sex, and batch (brain bank). The results were then compared to the meta-analysis results where a binomial test (sign test) was used to statistically evaluate consistency in direction of effect across the analyses.

**Reporting summary**

Further information on research design is available in the Nature Research Reporting Summary linked to this article.

## Data availability

The data supporting this study are available within the article, Supplementary information, or are publicly available. The BDR DNA methylation data have been deposited in the Dementias Platform UK (DPUK) data portal (https://portal.dementiasplatform.uk/CohortDirectory/Item?fingerPrintID=BDR) and the Gene Expression Omnibus (GEO) at accession number GSE197305.

## Code availability

Analysis scripts used in this manuscript are available on GitHub (https://github.com/gemmashireby/BDR_neuropathology_EWAS)[81]. Our detailed FANS protocol for the isolation of purified nuclei populations is available on protocols.io (https://doi.org/10.17504/protocols.io.bmh2k38e).

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

## Acknowledgements

G.S. was supported by a PhD studentship from the Alzheimer's Society. E.H., E.L.D., and J.M. were supported by Medical Research Council (MRC) grants K013807 and W004984 (awarded to J.M.). Data analysis was undertaken using high-performance computing supported by a Medical Research Council (MRC) Clinical Infrastructure award (M008924 awarded to J.M.). The analysis of FANS-purified nuclei was supported by Alzheimer's Research UK (ARUK) grant ARUK-PPG2018A-010 to E.L.D. DNA methylation data generated in the Brains for Dementia Research (BDR) cohort were supported by the Alzheimer's Society and Alzheimer's Research UK (ARUK). The BDR is jointly funded by Alzheimer's Research UK (ARUK) and the Alzheimer's Society in association with the Medical Research Council. The South West Dementia Brain Bank is part of the Brains for Dementia Research program, jointly funded by Alzheimer's Research UK (ARUK) and Alzheimer's Society, and is also supported by BRACE (Bristol Research into Alzheimer's and Care of the Elderly) and the Medical Research Council (MRC).

## Author contributions

J.B., S.P., B.C., and J.P.D., conducted laboratory experiments. G.S., E.H., R.G.S., E.P., and D.S.-V. undertook data analysis, bioinformatics, and/or support with data review. J.M., E.H., and E.L.D. conceived of the idea, obtained funding, and directed the project. S.L., A.T., P.F., K.B., and K.M. provided samples and clinical data. K.L. and R.G.S. provided data for meta-analysis. G.S., E.H., and J.M. drafted the manuscript. All authors read and approved the final submission.

## Competing interests

The authors declare no competing interests.

## Additional information

[1]Department of Clinical and Biomedical Sciences, Faculty of Health and Life Sciences, University of Exeter Medical School, University of Exeter, Exeter, UK. [2]Department of Psychiatry and Neuropsychology, School for Mental Health and Neuroscience (MHeNS), Maastricht University, Maastricht, The Netherlands. [3]Dementia Research Group, University of Bristol Medical School (Translational Health Sciences), Bristol, UK. [4]Translational and Clinical Research Institute, Newcastle University, Newcastle Upon Tyne, UK. [5]Biosciences, School of Science & Technology, Nottingham Trent University, Nottingham, UK. [6]Human Genetics, School of Life Sciences, University of Nottingham, Nottingham, UK. [7]Wolfson Centre for Age-Related Diseases, King's College London, London, UK. ✉e-mail: j.mill@exeter.ac.uk

