## [Peer Review File · Nature Communications]

DNA methylation signatures of Alzheimer's disease neuropathology in the cortex are primarily driven by variation in non-neuronal cell-typesREVIEWER COMMENTS

Reviewer #1 (Remarks to the Author):

Shireby and colleagues performed a systematic EWAS of AD neuropathology by profiling DNA methylation across >800,000 sites in two cortical brain regions (the dorsolateral prefrontal cortex [DLPFC] and occipital cortex [OCC]) in ~650 well-characterized donors from the Brains for Dementia Research (BDR) cohort. They further explored the extent to which AD-associated cortical differences in DNA methylation were driven by changes within specific cell populations.

The study offers a great resource for the AD community, but would benefit from some revision. Especially regarding the documentation of methods. In the current version of the manuscript any potential attempt to reproduce the results would be hampered by the lack of details. Below you can find the major and minor points to address.

Major points:

- One of the major shortcomings of the manuscript is associated with nomenclature/interpretation of their methods/results. Single nucleus RNA sequencing studies have shown that in an unbiased preparation approximately 15% of nuclei come from astrocytes and 5% come from microglia – meaning that in a NeuN-/SOX10- preparation ~2/3 of the nuclei would be of astrocyte origin while only ~1/3 would be of microglia. Based on the authors' own data, Figure S19 actually supports this, as in the double negative gate roughly 1/3 of the events are IRF8 positive (putative microglia marker) and approximately 2/3 are negative for this marker, and are most likely mainly astrocytes. Accordingly, to call the NeuN-/Sox- population “microglia” is not justified. Most fitting name would be “astrocyte enriched” but if the authors want to indicate there are microglia in there as well, the sample type could be called “astrocyte/microglia”.
- Figure S17 uses data from a couple of other DNA methylation studies, but only one of them is described (LBB1) - please describe the other two studies/datasets as well (MS, ROSMAP) and the analysis done on them
- please describe in detail in the methods section the computational method used to derive cell proportion estimates from DNA methylation data

Minor points:

- line 46 – purified neural cell types – microglia is not a neural cell type (same for line 113 and line 145)
- Figure S19 – please show full gating strategy (FSC/SSC, nuclear marker, doublet exclusion, etc) including the gating strategy for IRF8 staining
- Methods – please describe in detail the protocol used for the isolation of nuclei (tissue homogenization, staining, sorting, DNA extraction – including incubation times, concentrations, volumes, force used for centrifugation steps, etc.)
- Figures – please label each panel with the relevant information (e.g. brain region, pathology, specific gene investigated, etc.) so it would be easier to grasp the figures, as they are supposed to be self-explanatory
- Figure 4e – the categories are sample or cell types – they all come from the same tissue, so calling them "tissue types" is slightly misleading; same for Table 1

Reviewer #2 (Remarks to the Author):

The authors have identified novel and validated previously known epigenetic alterations in Alzheimer's disease cortex tissue. In contrast to most studies involving 'bulk' tissue, the authors were able to attribute most of their findings to non-neuronal cell subpopulation present in their samples.

The work is significant as it provides methylation profiles from 631 donors as well as methylation profiles of sorted neuronal cell populations on EPIC array platform. The authors identify 334 cortical differentially modified positions associated with AD pathology most of which are new. The authors also validated previously identified AD EWAS hits seen on older 450k arrays. Most importantly, the authors were able to attribute the changes to specific cellular populations found in 'bulk' cortex tissue. Finally, the analytical approach of testing multiple AD related covariates at once, deconvoluting signal from bulk tissue into cellular subtypes and performing meta-analysis with previously published data set the standard for further large scale EWAS studies.

The statistical analysis approaches are robust and sound. The data and the analysis performed were presented clearly. The following few aspects would improve the results:

- Please make raw IDAT files available in the GEO submission. Currently BDR DNA methylation data is only available upon application which hinders reproducibility of the findings.
- The methods used to process raw EPIC array data should be described in more detailed that would enable reproduction of the results. For example, `dasen()` function used to normalize the data takes the actual normalization method as an argument and it remains unclear what that method was.
- In addition to known confounders such as age and sex, PC1 was used to account for additional unexplained variance. However, when regression models are applied on two brain tissues, PC1 is most likely to represent tissue differences. Additional principal components should be used or methods that account for unknown and unwanted variability, e.g. SVA or RUV, could be employed.

Suggestion: accept with minor revisions

Reviewer #3 (Remarks to the Author):

This is a very interesting and informative investigation of brain tissue methylation differences according to different Alzheimer neuropathological indicators based on samples taken from almost 650 individual brains. The study includes a) a EWAS across >800,000 sites in the dorsolateral prefrontal cortex [DLPFC] and occipital cortex [OCC] b) a meta-analysis of the newly generated data with a previous AD EWAS meta-analysis for more than 2,000 individuals; and c) DNA methylation analyses conducted in purified nuclei populations enriched for neurons, oligodendrocytes and microglia for small subset of donors with low versus high AD pathologies. This is a well conducted study that uses state-of the science approaches and the manuscript is well-written, concise, and informative.

Apart from finding some novel DNAm variation at loci not previously implicated in dementia (as the authors used the larger EPIC array and increased their statistical power in meta-analysis), one of the most interesting results the authors present is that AD-associated methylomic variation in the cortex primarily reflects differences in microglia methylation with only about 5% of DMPs identified in the EWAS being attributed to neuronal cell types according to cell-sorted analyses. The results suggested that the differences in methylation were not due to cell type composition differences or to neuronal loss with increasing pathology but were indeed mainly driven by methylation changes in the microglial subset of cell types. It is not surprising that the pathway analyses indicated inflammation and immune cell pathways in addition to some brain development related pathways as being enriched.

Overall the results of this paper make an important contribution to the literature providing novel data on methylation in brain tissues in those affected with AD and related types of dementia based on

detailed pathologies. Unfortunately, the authors only interpret their results in very general terms referencing inflammation and neurodevelopmental processes, leaving the reader wondering what it means that they identified 334 cortical DMPs associated with AD pathology with a number of them being novel and how this information can be helpful or employed in the future. If the authors have thought about this, presenting these thoughts would be of interest to the reader.

Please consider these questions/ comments:

- 1) The authors are providing pathway analysis results only in a supplementary table; and describe the results in one sentence. However, these results might be of greater interest and the authors should consider presenting a figure or image that summarizes these results for the reader.
- 2) On page 9, the authors point out that "PDLIM2 has been implicated in PD by GWAS" and also state that this "one DMP (cg18100976, annotated to PDLIM2) was identified in both the DLPFC and OCC" – however it is unclear whether this or any of the other 67 AD pathology DMPs found to be 'highly consistent in analyses of non-AD (Braak LB Stage and TDP-43 status) neuropathology measures" included this gene or not. The authors refer the reader to Supplementary Figure S12 but this figure does not allow us to identify the coefficients for any one of these loci and the tables S2 also does not contain the non-AD effect size information. Please add this information to Table S2.
- 3) On page 14, the authors state that " we cannot exclude the possibility that the differences identified reflect the influence of other factors related to AD pathology that were not controlled for in this study, for example environmental factors such as medication exposure." Wanting to control for 'other factors' generally implies that the authors consider these factors confounders. However, it is more likely that medication is affected by the pathology i.e. follows the pathologic changes and clinical phenotype rather than affects or changes the AD pathology. Thus, medication would be considered an intermediate factor in the pathway to methylation changes (if pathology is causing methylation differences or is correlated with them through medication use) and not a confounder. I like to encourage the authors to clarify this point.
- 4) It is unclear how this statement on page 11 lines 340-43 "These results suggest that the widespread cortical DNA methylation differences associated with AD neuropathology are primarily manifest in non-neuronal cell-types, although there is evidence for pathology associated differences in DNA methylation in neuronal cell types for a subset (5%) of DMPs (Supplementary Table S14)." is indeed supported by the data provided in Table S14.
- 5) The table S4 seems mislabeled as supplementary table S6 – please check the supplementary table labels carefully

NCOMMS-22-12802-T- Response to reviewer comments

Following the highly positive comments from all three Reviewers we are delighted that we have been given the opportunity to submit a revised version of our manuscript to *Nature Communications*. Reviewer 1 concludes that "...*The study offers a great resource for the AD community...*". Reviewer 2 concludes that our "...*work is significant...*" and that our "...*statistical analysis approaches are robust and sound...*". Reviewer 3 concludes our manuscript is "...*a very interesting and informative investigation...*" and that our results "...*make an important contribution to the literature...*".

We would like to thank each of the Reviewers for the detailed and constructive comments on our original submission, and we appreciate their suggestions for improvements on the manuscript. Please find attached our revised manuscript with changes highlighted.

Below we address the specific comments from each of the Reviewers.

Reviewer 1

Major comments

- 1. One of the major shortcomings of the manuscript is associated with nomenclature/interpretation of their methods/results. Single nucleus RNA sequencing studies have shown that in an unbiased preparation approximately 15% of nuclei come from astrocytes and 5% come from microglia – meaning that in a NeuN-/SOX10- preparation ~2/3 of the nuclei would be of astrocyte origin while only ~1/3 would be of microglia. Based on the authors' own data, Figure S19 actually supports this, as in the double negative gate roughly 1/3 of the events are IRF8 positive (putative microglia marker) and approximately 2/3 are negative for this marker, and are most likely mainly astrocytes. Accordingly, to call the NeuN-/Sox- population "microglia" is not justified. Most fitting name would be "astrocyte enriched" but if the authors want to indicate there are microglia in there as well, the sample type could be called "astrocyte/microglia".**

We agree that the double-negative population is likely to be more heterogeneous than the two positively sorted populations (NeuN+ and SOX10+). Our initial terminology for this fraction ('microglia-enriched') was intended to indicate that the proportion of microglial nuclei in this fraction is enriched compared to bulk/total nuclei (and the two positively-sorted populations). The Reviewer is correct, however, that there are also other glial and non-neural cells (especially astrocytes) in the NeuN-/SOX10- population and we agree that our nomenclature is confusing. Of note, when performing our FANS experiments we also co-stained nuclei with the microglial marker IRF8. IRF8+ nuclei were found to show complete overlap with the NeuN-/SOX10- population, confirming the efficiency of our sorting protocol. In response to the Reviewer's comment, we have gone back to our raw FANS data and empirically calculated the proportion of IRF8+ (microglial) nuclei in the double-negative population. On average a large proportion of NeuN-/SOX10- nuclei were IRF8+ (mean = 42.23%) indicating a relatively strong enrichment of microglia in the double-negative population. However, we agree that the term 'microglia-enriched' is potentially misleading so have instead renamed each population based on the actual gating strategy used – i.e. NeuN+, SOX10+ and NeuN-/SOX10-. This has been updated in the main text, all figures and tables.

2. **Figure S17 uses data from a couple of other DNA methylation studies, but only one of them is described (LBB1) - please describe the other two studies/datasets as well (MS, ROSMAP) and the analysis done on them.**

We thank the Reviewer for highlighting this point and apologize that our description of the cohorts included in the previous meta-analysis by Smith et al was not complete. We have now added a sentence to the legend of this figure (in addition to Figure 3) describing the specific cohort each abbreviation relates to. Furthermore, we have clarified in the text that the Smith et al meta-analysis includes data from multiple cohorts and that a full description of each is provided in the manuscript. We also include a Supplementary Table which has the numbers included from each of the individual cohorts (**Supplementary Table S7**).

3. **Please describe in detail in the methods section the computational method used to derive cell proportion estimates from DNA methylation data**

We have included additional information about the cell deconvolution method in the **Methods** section as requested. Briefly, this method combines the cell-type reference data (generated from FANS-isolated nuclei populations) with the bulk cortex data and performs quantile normalization. It then performs an ANOVA to identify sites that are significantly different ($P < 1 \times 10^{-8}$) between the different cell types, selecting 100 sites per cell type (50 hypermethylated and 50 hypomethylated). These sites are then used to derive cellular proportions using quadratic programming, in essence a least squares minimization, with the constraint that all the proportions are greater than or equal to 0 and the sum of the three proportions is less than or equal to 1.

Minor Comments

1. **line 46 – purified neural cell types – microglia is not a neural cell type (same for line 113 and line 145)**

We have now amended this throughout the manuscript.

2. **Figure S19 – please show full gating strategy (FSC/SSC, nuclear marker, doublet exclusion, etc) including the gating strategy for IRF8 staining**

We have replaced Figure S19 (now numbered Figure S20) with a new figure (see **Figure 1** below) that clearly shows the full FANS gating strategy. The figure highlights the IRF8+ population and its overlap with the NeuN-/SOX10- population.

Figure 1 (new Figure S20): Isolation of cell-type-enriched nuclei populations from DLPFC tissue using fluorescence activated nuclei sorting (FANS). Shown is the FANS gating strategy for one representative BDR DLPFC sample highlighting the isolation of three discrete nuclei populations (NeuN+, SOX10+, NeuN-/SOX10-) and the overlap of nuclei positive for IRF8, a microglial marker, with the NeuN-/SOX10- population.

3. Methods – please describe in detail the protocol used for the isolation of nuclei (tissue homogenization, staining, sorting, DNA extraction – including incubation times, concentrations, volumes, force used for centrifugation steps, etc.)

We have added some additional information to the Methods section. Including the full step-by-step protocol for our FANS isolation protocol would take several pages of text and be too much detail for a manuscript. However, the full detailed protocol is posted on protocols.io (<https://dx.doi.org/10.17504/protocols.io.bmh2k38e>) and this is now more explicitly referenced in the manuscript.

4. Figures – please label each panel with the relevant information (e.g. brain region, pathology, specific gene investigated, etc.) so it would be easier to grasp the figures, as they are supposed to be self-explanatory

Where this information improves understanding of a figure (or panel) we have now included relevant labels as requested. In cases where specific labels have not been added, we have included all information in the figure legend.

5. **Figure 4e – the categories are sample or cell types – they all come from the same tissue, so calling them "tissue types" is slightly misleading; same for Table 1**

This has been changed as requested.

Reviewer 2

1. **Please make raw IDAT files available in the GEO submission.**

To clarify, our raw data are all available on GEO and also from the DPUK data portal. This is clearly stated in the Data Availability section of the manuscript.

2. **The methods used to process raw EPIC array data should be described in more detailed that would enable reproduction of the results. For example, dasen() function used to normalize the data takes the actual normalization method as an argument and it remains unclear what that method was.**

We have now clarified in the Methods section that the default options were used.

3. **In addition to known confounders such as age and sex, PC1 was used to account for additional unexplained variance. However, when regression models are applied on two brain tissues, PC1 is most likely to represent tissue differences. Additional principal components should be used or methods that account for unknown and unwanted variability, e.g. SVA or RUV, could be employed.**

We ran a series of analyses to establish the optimal EWAS model to use on the BDR DNA methylation data. We first performed principal component (PC) analysis finding that PC1 explained almost all variation in the DNA methylation data (96.8%), with subsequent PC's explaining only a very small proportion of the variation (e.g. PC2 = 0.76%, PC3 = 0.34%, PC4 = 0.23%, PC5 = 0.15%). We then tested for an association between PC1 and all known traits to identify how much of the variation is explained by available covariates. Experimental plate (i.e. batch) explained nearly half of the variation in PC1 (44.94%), and with the addition of cell proportions this increased to ~60%. Further available covariates had little influence on PC1; it is worth noting given the Reviewers' comment that 'brain region' had a non-significant correlation with PC1, explaining only 0.43% of the variance once cell-type proportions are accounted for. Therefore, the inclusion of PC1 is unlikely to represent differences between the two cortical regions profiled in this study. We therefore decided to include PC1 as a covariate in our analyses of neuropathology to account for unknown variation, with other known variables being added as specific confounders. Of note, although some of the known confounders are correlated with PC1, the correlations are small enough that co-linearity should not be an issue in the model.

The Reviewer suggested also performing surrogate variable analysis (SVA); this is something we had already undertaken in the course of our analyses but not included in the manuscript. Nearly all the variation in PC1 was explained by SV1 and SV2 combined ($R^2 = 95.52$), with additional SVs only explaining a very small proportion of extra variance. Although SVs are generally hypothesized to account for unexplained variation, we investigated which measured traits, if any, explained the variation in each SV. SV1 was predominantly explained by experimental batch (plate) and derived cell proportions; in combination these variables explained 97.16% of the variation in SV1. SV2 was also

predominantly explained by plate and derived cell proportions ($R^2 = 67.29\%$), however around 30% of the variation in SV2 was not explained by known covariates. Our comparisons show that a greater number of SVs than PCs would be required in a regression model to explain unknown variance in the data. Therefore, in order to ensure our model was parsimonious we opted to include only PC1 in the regression models reported in the manuscript. Of note, only a very small amount of the variance in PC1 is explained by the target phenotypes (i.e. neuropathology) of our EWAS analysis and it captures more of the unknown variance than SVs without hampering the degrees of freedom.

Reviewer 3

1. The authors are providing pathway analysis results only in a supplementary table; and describe the results in one sentence. However, these results might be of greater interest and the authors should consider presenting a figure or image that summarizes these results for the reader.

We have now included a supplementary figure which visualises the GO results (see new **Supplementary Figure S17**).

2. On page 9, the authors point out that “PDLIM2 has been implicated in PD by GWAS” and also state that this “one DMP (cg18100976, annotated to PDLIM2) was identified in both the DLPFC and OCC” – however it is unclear whether this or any of the other 67 AD pathology DMPs found to be ‘highly consistent in analyses of non-AD (Braak LB Stage and TDP-43 status) neuropathology measures’ included this gene or not. The authors refer the reader to Supplementary Figure S12 but this figure does not allow us to identify the coefficients for any one of these loci and the tables S2 also does not contain the non-AD effect size information. Please add this information to Table S2.

As requested, we have now updated **Supplementary Table S2** to include the summary statistics for the two non-AD neuropathology measures so that differences at individual DNA methylation sites can be explored. Given the relatively small numbers for non-AD pathology we were not well-powered to definitively assess consistency, but overall effect sizes for both Braak LB stage and TDP-43 status were correlated with those for AD pathology.

3. On page 14, the authors state that “ we cannot exclude the possibility that the differences identified reflect the influence of other factors related to AD pathology that were not controlled for in this study, for example environmental factors such as medication exposure.” Wanting to control for ‘other factors’ generally implies that the authors consider these factors confounders. However, it is more likely that medication is affected by the pathology i.e. follows the pathologic changes and clinical phenotype rather than affects or changes the AD pathology. Thus, medication would be considered an intermediate factor in the pathway to methylation changes (if pathology is causing methylation differences or is correlated with them though medication use) and not a confounder. I like to encourage the authors to clarify this point.

We apologise if this was not clear. We were trying to highlight that the changes we observe might not actually be directly associated with the accumulation of AD pathology but be induced by other factors (such as medication) that individuals with AD pathology are

exposed to. Although we agree that this is not therefore a confounder in the way that certain demographic factors might be, it still makes determining cause vs effect difficult. We believe it is important to be conservative in interpreting DNA methylation data, but have reworded the sentence so that it is hopefully clearer.

- 4. It is unclear how this statement on page 11 lines 340-43 "These results suggest that the widespread cortical DNA methylation differences associated with AD neuropathology are primarily manifest in non-neuronal cell-types, although there is evidence for pathology associated differences in DNA methylation in neuronal cell types for a subset (5%) of DMPs (Supplementary Table S14)." is supported by the data provided in Table S14.**

We agree this sentence was not clear. We have now clarified in the Results that there are 27 (8.3%) of DMPs at which effect sizes identified in bulk cortex are positively correlated with differences in NeuN+ nuclei - but not other nuclei populations - suggesting that these might be neuron-specific differences.

- 5. The table S4 seems mislabelled as supplementary table S6 – please check the supplementary table labels carefully**

We thank the Reviewer for highlighting the mislabelling which has now been amended.

REVIEWER COMMENTS

Reviewer #2 (Remarks to the Author):

Thank you for the detailed responses. I have no further comments.